# Barriers and facilitators of adherence to social distancing recommendations during COVID-19 among a large international sample of adults

Adina Coroiu[1]*, Chelsea Moran[2], Tavis Campbell[2], Alan C. Geller[1]

**1** Department of Social and Behavioral Sciences, Harvard T.H. Chan School of Public Health, Boston, Massachusetts, United States of America, **2** Department of Psychology, University of Calgary, Calgary, Alberta, Canada

\* acoroiu@hsph.harvard.edu

## Abstract

### Background

Social distancing measures (e.g., avoiding travel, limiting physical contact with people outside of one's household, and maintaining a 1 or 2-metre distance between self and others when in public, depending on the country) are the primary strategies used to prevent transmission of the SARS-Cov-2 virus that causes COVID-19. Given that there is no effective treatment or vaccine for COVID-19, it is important to identify barriers and facilitators to adherence to social distancing to inform ongoing and future public health campaigns.

### Method

This cross-sectional study was conducted online with a convenience sample of English-speaking adults. The survey was administered over the course of three weeks (March 30 – April 16, 2020) when social distancing measures were well-enforced in North America and Europe. Participants were asked to complete measures assessing socio-demographic characteristics, psychological constructs, including motivations to engage in social distancing, prosocial attitudes, distress, and social distancing behaviors. Descriptive (mean, standard deviation, percentage) and inferential statistics (logistic regression) were used to describes endorsement rates for various motivations, rates of adherence to social distancing recommendations, and predictors of adherence.

### Results

Data were collected from 2013 adults living primarily in North America and Europe. Most frequently endorsed motivations to engage in social distancing (or facilitators) included "I want to protect others" (86%), "I want to protect myself" (84%), and I feel a sense of responsibility to protect our community" (84%). Most frequently endorsed motivations against social distancing (or barriers) included "There are many people walking on the streets in my area" (31%), "I have friends or family who need me to run errands for them" (25%), "I don't trust

**Funding:** AC is supported by post-doctoral research fellowships from the Canadian Institutes of Health Research (CIHR) and Fonds de Recherche du Quebec – Santé (FRQS). CM is supported by a Vanier Canada Graduate Scholarship and a University of Calgary Training in Research and Clinical Trials in Integrative Oncology (TRACTION) fellowship.

**Competing interests:** The authors have declared that no competing interests exist.

the messages my government provides about the pandemic" (13%), and "I feel stressed when I am alone or in isolation" (13%). Adherence to social distancing recommendations ranged from 45% for "working from home or remotely" to 90% for "avoiding crowded places/non-essential travel", with men and younger individuals (18–24 years) showing lower adherence compared to women and older individuals.

## Conclusion

This study found that adherence to social distancing recommendations vary depending on the behaviour, with none of the surveyed behaviours showing perfect adherence. Strongest facilitators included wanting to protect the self, feeling a responsibility to protect the community, and being able to work/study remotely; strongest barriers included having friends or family who needed help with running errands and socializing in order to avoid feeling lonely. Future interventions to improve adherence to social distancing measures should couple individual-level strategies targeting key barriers to social distancing identified herein, with effective institutional measures and public health interventions. Public health campaigns should continue to highlight compassionate attitudes towards social distancing.

## Background

The incidence of SARS-Cov-2 virus, which causes the disease called COVID-19, has increased dramatically worldwide since December 2019, when the first case was recorded in humans [1] Currently, no effective pharmaceutical treatment or vaccine exist. It is believed that SARS-Cov-2 can be transmitted by both symptomatic and asymptomatic individuals [2–4] and its rate of transmission is higher than that of the influenza virus [5, 6], which makes it highly contagious. Since the World Health Organization (WHO) declared COVID-19 a pandemic on March 11, 2020, national and international public health agencies proposed several measures to contain or mitigate the virus transmission. In Canada, the United States, and some European countries, these range from virus containment strategies (e.g., complete quarantine of the population of an entire region, as in Wuhan, China) to mitigation of transmission through various degrees of measures designed to keep physical distance between individuals (i.e., social/physical distancing), coupled with rigorous personal hygiene (e.g., washing hands frequently and thoroughly, avoiding touching the eyes, nose, and mouth, coughing and sneezing into the elbow) and wearing face masks when in public.

For countries adopting a "mitigation scenario" social distancing measures, including avoiding travel, limiting physical contact with people outside of one's household, and maintaining a 2-metre distance between self and others when in public, are the primary strategies used to prevent the over-burdening of health care systems by reducing the rate of transmission at the level of the general population [7, 8]. More stringent measures, including full quarantine and isolation have been recommended for individuals at high risk for contracting the virus, such as older individuals and those with pre-existing medical conditions [9, 10]. Prediction modelling investigating various scenarios of prolonged and intermittent social distancing, suggested some form of these measures may be required into 2024 to prevent overloading of health care systems, absent effective therapeutic interventions and accurate knowledge of immunity duration for those infected with SARS-Cov-2 [11]. These results are also supported by other analyses, currently in pre-print (i.e., not yet gone through peer review), suggesting multiple or extended periods of social distancing might be needed in the future [12–16]. However, many

modelling estimates assume high compliance to public health measures by the general population [17], which may not adequately represent actual practice of health behaviours, such as social distancing.

Given that social distancing measures ("stay-at-home" or "shelter-in-place" orders) are imposing significant lifestyle changes for the general population and they may potentially be required for months or years to come, it is important to understand what facilitates or prevents adherence to these measures, so that public health interventions could be developed in a timely manner. Because most countries have relaxed their social and physical distancing measures compared to the measures taken in the early days of the epidemic, it is crucially important to determine the factors that might affect adherence to these preventive health behaviours in the long run. Behavioural and social scientists are well positioned to help answer these questions and help guide COVID-19 prevention interventions, by incorporating messaging that targets a shared sense of identity, norms of prosocial behavior, emphasize benefits to the recipient, focus on protecting others or each other, align with the recipient's moral values, appeal to social consensus or scientific norms, highlight the prospect of social group approval; avoid fear-based messages or those inducing disgust towards other people, or avoid authoritarian messages, such as those based on coercion [18, 19]. Emerging pre-publication literature assessing the best strategies to facilitate adherence to COVID-19 preventive measures found that prosocial framing of the preventative message (i.e., "don't spread it"; benefit to others) was more effective than personal/self-interest framing (i.e., "don't get it"; benefit to self) in sample of 988 people recruited from the United States in mid-March 2020 [20]. Further, in an experimental within-subjects study of 955 people from the United States, information presented using both threatening language and prosocial language, the latter condition had larger effects on compliance when associated with highly positive emotional responses [21]. Prosocial framing was also associated with increased intentions to engage in social distancing and proper hygiene behaviours [20] social isolation [21]. Further, a series of four experimental studies investigating the role of prosocial emotions in motivating COVID-19 preventive behaviours (total N = 3,718 from Germany, USA and UK) found that empathy for those vulnerable to the virus was associated with increased social distancing behaviors and inducing empathy promoted motivation to adhere to COVID-19 preventative measures [22]. Lastly, an experimental study with two active conditions and a control in a sample 500 people recruited from Ireland found that messages that highlighted the risk of infecting vulnerable people or the risk of infecting large numbers of people led to increased intentions to engage in social distancing behaviours and increased acceptability of these behaviours [23].

Various models of health behaviour change conceptualize motivation as a central predictor for the adoption and maintenance of preventative health behaviours. For example, the Capability-Opportunity-Motivation-Behaviour (COM-B) model [24] posits that the interaction between individual capability (or having the necessary knowledge and skills) and opportunity (physical, social, and environmental support) directly influence motivation to engage in a behavior (reflective and automatic processes driving behavior), which leads to behaviour change and maintenance. Self-determination theory [25] suggests there are two types of motivations that drive behaviour change, intrinsic motivation, where the individual derives pleasure from the behavior, and extrinsic motivation, where external pressures are facilitating adherence to behaviour. Lastly, Motivational Interviewing [26], which is primarily an intervention modality, posits that motivations are the driving force for behavioral change, and are reflected in personal statements closely related to core values.

In the context of the COVID-19 pandemic, it seems reasonable to assume that motivations or individual reasons to adhere to recommendations about social distancing (e.g., desire to protect self and others) as well as external circumstances or motivators (e.g., workplace/school

conducted remotely) contribute to engagement in and adherence to preventative behaviours, such as social distancing. These motivations also likely interact with various sociodemographic variables, such as gender, age, socioeconomic and minority status, health status, and household size and composition. For instance, it has been found that mortality from and severity of COVID-19 is higher among men [27–29], older individuals [28, 30], those with predisposing conditions [30], and racial minorities. In a Southern state of the United States with a population of 31% Black, hospitalization rate was 77% and mortality rate was 71% for Blacks compared to whites [31]. Socioeconomic status also intersects with size of the household, with economically disadvantaged individuals being more likely to live in overcrowded housing, limiting the ability to socially distance [32]. Further, health status of other individuals in the household (e.g., living with family members that are more vulnerable to COVID-19 infection and health consequences such as older people or individuals with pre-existing health conditions) may also have an impact on motivation and social distancing behaviour.

### Research aims

This study has three aims:

1. to describe rates of motivations (barriers and facilitators) for social distancing;

2. to describe rates of adherence to social distancing recommendations;

3. to investigate the relationship between socio-demographic characteristics, psychological variables, and motivations for social distancing *and* adherence to social distancing recommendations among a large, convenience sample of English-speaking adults recruited primarily from Europe and North America.

## Methods

### Study design

This study used a cross-sectional survey design. Recruitment and data collection were conducted online using the Qualtrics platform. Ethical approval was obtained from the University of Calgary Conjoint Health Research Ethics Board. The reporting of the study followed the STROBE guideline [33].

### Participants and procedures

The survey was hosted on the Qualtrics platform and was distributed via snowball convenience sampling through co-author's professional and personal networks and social media accounts (e.g., Twitter, Facebook); ads posted on University of Calgary online platforms; via paid ads (35.00 CAD/day) posted on Facebook targeting English-speaking adults residing in North America and Europe. Data collection was conducted between March 29th, 2020 and April 16, 2020, when strict regulations about social distancing were in place in North America and most countries in Europe. Paid targeted Facebook ads were placed between April 1st and April 12th, 2020. A preliminary version of the online survey was piloted on 15 individuals whose data were not included in this report and who provided edits to the items to improve readability, suggestions for the question flow, and corrections for small grammatical errors. No identifiable data (name, contact information, IPs) were collected through the survey. A copy of the final version of the survey can be found at https://doi.org/10.17605/OSF.IO/YX67C.

Eligibility criteria for this study included being 18 years of age or older, ability to read and write in English, and having access to the internet. Participants provided informed consent

online, by clicking on a bullet, indicating that they had read through and understood the conditions of their participation in this study. The survey included questions about socio-demographic and medical variables, psychological constructs, including motivations for social distancing, and social distancing behaviors. The average time for completion of the survey was 20.36 minutes (SD = 99.26), and 75% of the sample completed the survey in less than 16 minutes (25th percentile: 9.5; minutes; 50th percentile: 12.2; 75th percentile: 16.45 minutes).

**Patient and public involvement.** Aside from providing data for this study, participants were not involved in any other aspect of this research project.

## Predictor variables

**Sociodemographic and medical information.** Participants were asked to indicate their age, gender, highest level of education completed, country of residence, whether they had a medical condition associated with an increased risk for COVID-19, self-perceived symptoms of COVID-19 over the previous week, and whether they were tested for COVID-19.

**MacArthur scale of subjective social status scale [34].** Perceived socioeconomic status (SES) was assessed via a single item consisting of a picture of a 10-step ladder. Participants were asked to select the rung of the ladder that best represents their socioeconomic position related to others in society, where higher scores indicate higher perceived SES. In a sample of white women, this measure was associated with income, education, and self-rated health status [34]. This measure demonstrated adequate test-retest reliability (ICC = .67, weighted kappa statistic = .62) in a general population sample from Brazil [35].

**Health literacy scale.** Health literacy was assessed with one item created by the study authors "If I was given a pamphlet on how to prevent a medical condition (disease), I would be able to understand the main message(s)", with response options on a 4-point Likert scale ranging from 1 (strongly disagree) to 4 (strongly agree).

**Belief in conspiracy theories scale [36].** Conspiracy beliefs were assessed using a single-item measure, which provides a scenario about common conspiracy theories and asked respondents to rate the following statement using a Likert-type scale ranging from 1 ("completely false") to 9 ("completely true"): "I think that the official version of the events given by the authorities very often hides the truth". The scale showed good predictive validity, test-retest reliability, and convergent validity with lengthier scales assessing the same construct in samples of students and MTurk workers [36].

**Prosocial behavioral intentions scale [37].** Prosocial attitudes were assessed with the 4-item scale inquiring about participants' willingness to perform prosocial behaviours on an average day (sample item, "comfort someone I know after they experience hardship). Answers were scored on a 7-point Likert scale ranging from 1 ("I wouldn't do this") to 7 ("I would do this"). Total scores ranged from 4 to 28, with higher scores indicating more positive attitudes towards prosocial behaviour. The scale was associated with past prosocial behaviour and measures of morality in a general population sample recruited via MTurk [37]. Cronbach's alpha was .81 in the validation sample and .76 in the current sample.

**Patient health questionnaire-4 (PHQ-4) [38].** Psychological distress, conceptualized as symptoms of anxiety (sample item, "Not being able to stop or control worrying") and depression (sample item, "Little interest or pleasure in doing things"), was assessed using the PhQ-4. Respondents were asked to indicate whether they experienced symptoms over the previous two weeks using a 4-point Likert scale ranging from 0 ("not at all") to 3 ("nearly every day"). Total scores ranged from 0 to 12, with higher scores indicating higher distress. In a large general population sample the scale was found to be valid and reliable when compared to longer

symptom inventories assessing anxiety and depression [39]. Cronbach's alpha for the PHQ-4 was .78 in the validation sample .87 in the current sample.

**Motivations for social distancing.** A Motivational Interviewing framework was used to conceptualize personal motivations regarding social distancing and isolation recommendations [26, 40]. This approach suggests that motivations for and against behaviour change and adherence can exist simultaneously within an individual, and that encouraging individuals to express what motivates them to adopt a certain behaviour helps highlight ambivalence toward health behaviour change. Additionally, the social-ecological model [41] was used to organize a set of 55 statements reflecting various motivations related to adherence to social distancing, created through team discussions including all study authors and piloted with a sample of 15 participants and subsequently revised for clarity and consistency. These motivations were classified as individual (n = 20), interpersonal (n = 12), organizational (n = 9), or community (n = 14) related motivations. Further, each cluster included motivations that were conceptualized by study authors as being motivations for or against adherence to social distancing behaviours. For example, at the individual-level, "I want to protect myself" was generated as a motivation that supports adherence to social distancing behaviour (i.e., motivation "for") while "I feel stressed when I'm alone or in isolation" was identified as a motivation that could act as a barrier to social distancing behaviour (i.e., motivation "against"). A complete list of motivation items, as circumscribed to the levels of the social-ecological model, is included in Table 2 of the results. A graphical representation of the social-ecological model was included in (Fig 1).

Participants were given the following instructions: "Below is a list of motivations for social distancing. Some may be reasons to follow social distancing rules and others may be reasons that may make you skeptical or hesitant about following social distancing rules. Please indicate which of the statements below reflect your motivations for social distancing, by selecting all that apply." For data analyses, endorsed items were coded as "1" and non-endorsed items as "0".

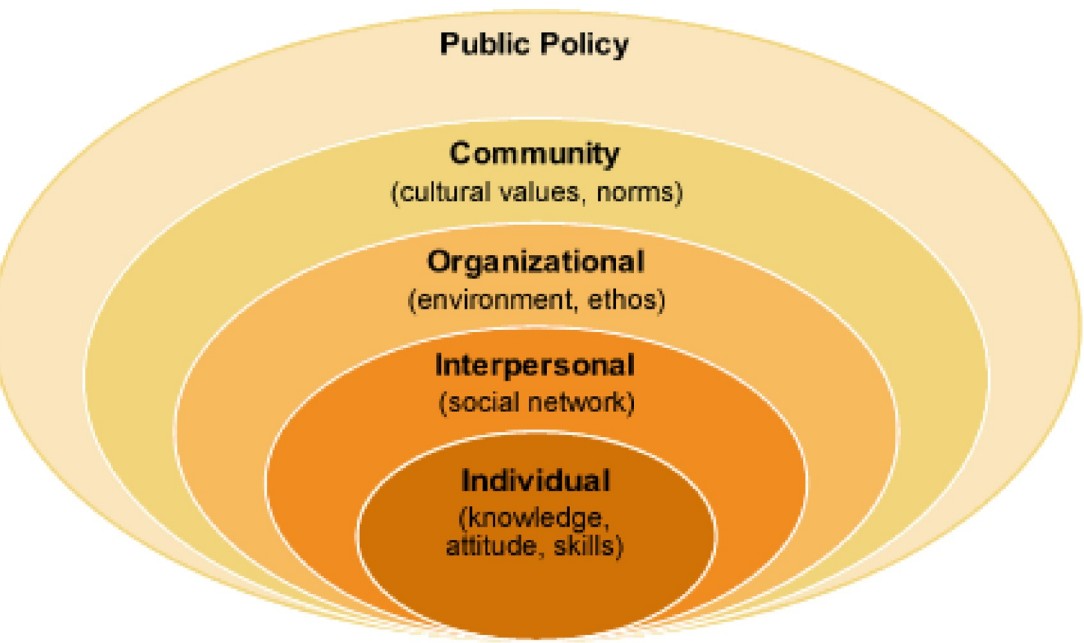

**Fig 1. Social-ecological model.** Source: McLeroy et al. [41].

## Outcome variables

**Social distancing behaviours.** A list of 15 behaviours consistent with social distancing recommendations from national and international public health authorities [42–44] was generated by the study team. The term "social distancing" was used in favour of the term "physical distancing" because this was the term most commonly used by public health agencies and the media at the time of study conceptualization. Items included references to working from home, practicing social distance from various groups, avoiding large social gatherings or travel, keeping a safe distance from others when in public, and isolating at home when sick.

Participants were asked to rate each behaviour according to the following prompt: "Please indicate to what extent have you done any of these behaviours in the past week (7 days, including today)". Response options ranged from 1 ("Never") to 4 ("Always"). Items also included a 'Not applicable' option to account for the possibility that not all social distancing behaviours are necessary or relevant for all (e.g., working from home for jobs that cannot be completed remotely or completing coursework remotely for non-students). Adherence to social distancing was conceptualized as "always" endorsing the behaviour (coded as "1") whereas non-adherence was conceptualized as behaviour endorsed less often than "always", including "never", "sometimes", or "often" response choices (coded as "0"). This dichotomy was created based on conceptual reasons, given that social distancing is effective only when practiced consistently.

## Data analysis

Descriptive analyses (%, M/SD) were computed for all study measures. Chi square tests were used to compare adherence to social distancing behaviors, obtained in the first, second and third week of recruitment. Logistic regression analyses were used to test the association between socio-demographic (age, gender, education, country of residence, medical status, and COVID-19 symptoms), psychological (conspiracy beliefs, health literacy, prosocial behavior, distress), and motivational predictors and social distancing behavioral outcomes. During data collection, recommendations and policies for social distancing differed by region or country but did not change within one region or country, hence our regression models did not account for timing of survey completion.

Post hoc decisions for the selection of motivation and social distancing items to be included in the logistic regressions included a) identifying two motivations "for" (or facilitators) and two "against" (or barriers) social distancing with the highest endorsement rates from each of the four clusters of motivations; and b) excluding social distancing items with adherence rates of > 85% (high) or < 10% (low) as well as items with > 45% "not applicable" answers. In regressions analyses, the variable country of residence was dichotomized into countries with strictly enforced guidelines for social distancing i.e., lockdown enforced by government authorities or police, versus countries with recommended guidelines for social distancing, but not enforced by government organizations or police. The coding sheet we used to collect information and code data about national policies about social distancing is available at https://osf.io/yx67c/.

## Results

Data were collected from N = 2336 participants, of whom 14.8% completed less than 40% of the questionnaire and were thus excluded. Analyses were conducted with N = 2013 individuals who provided answers to > = 60% of questions with 100% completion rate for the outcome measure, social distancing behaviors. Respondents completed the survey over three weeks, as follows: first week, March 29-April 4, n = 635 (31.5%); second week, April 5–11, n = 900 (44.7%); third week, April 12–16, n = 478 (23.7%).

## Sample characteristics

Among the entire sample (n = 2013), 84% were female, 71% had completed at least a bachelor degree, 38.8% resided in North America (Canada and the United States) versus 59.5% in Europe versus 1.7% other locations; 30.9% had a pre-existing medical condition that made them vulnerable to COVID-19, 25% had experienced at least one symptom associated with COVID-19, and 3% had been tested for COVID-19. With respect to psychological variables, participants endorsed average conspiracy beliefs, health literacy, and distress levels, and increased prosocial attitudes. Detailed descriptive statistics for the study measures are included in Table 1.

## Motivations for social distancing

Endorsement rates of motivations "for" (*facilitators*) and "against" (*barriers*) social distancing behaviours are included in Table 2, organized according to four levels of the social-ecological model.

 **Facilitators of social distancing.** Top two most endorsed individual-level facilitators included "I want to protect myself" (84%) and "I want to avoid spreading the virus to others" (83%); interpersonal factors included "I want to protect others" (86%) and "I feel a sense of responsibility to protect our community" (84%), organizational-level factors included "my workplace/ school recommended we practice social distancing" (54%) and "my workplace /school conducts operations remotely" (51%); and community-level factors included "restaurants in my area are closed for eating-in" (95%) and "community centers and recreational facilities in my area are closed" (94%).

 **Barriers to social distancing.** Top two most endorsed individual-level barriers included "I don't trust the messages my government provides about the pandemic (13%), and "I feel stressed when I am alone or in isolation" (13%); interpersonal barriers included "I have friends or family who need me to run errands for them" (25%) and "I socialize with people to avoid feeling lonely" (6%); organizational-level barriers included "my work cannot be done remotely" (16%) and "my workplace requires me to come into work" (11%); and community-level barriers included "There are many people walking on the streets in my area" (31%) and "It is not possible to shop online and get deliveries in my area" (11%).

 Of importance, least endorsed individual-level barriers included "I believe the best strategy to manage this pandemic is to let the virus run its course" (3.9%), "I think the government is exaggerating the impact of this pandemic" (3.8%), "I think I cannot spread the virus if I am not sick" (1.9%), and "I've heard social distancing is not effective at reducing transmission of the virus" (1.1%). Least endorsed interpersonal barriers included "I believe my actions cannot protect others from contracting the virus" (3.2%) and "I believe it's OK to go out and meet with people in small groups" (2.9%).

## Adherence to social distancing behaviors

Detailed descriptive statistics for the social distancing behaviors are included in Table 3. Rates of social distancing behaviours varied slightly across the three weeks of recruitment (Table 3). There was no perfect adherence (100%) for any of the social distancing behaviours assessed. Adherence > = 90% was found for avoiding crowded places. Adherence in the 80–89% range was found for avoiding non-essential gatherings, avoiding going out to places, avoiding close-contact greetings, avoiding contact with high risk people, and avoiding seeing friends in person. Adherence to keeping a 2-meter distance from others was endorsed by 66% and staying at home when sick was endorsed by 46%. Lowest endorsement rates (6–8%) were reported for behaviours related to ordering take-out and getting food delivered.

**Table 1. Sample characteristics (N = 2013).**

| Variable | N | n (%) | M (SD) | Range |
|---|---|---|---|---|
| Age | 2013 | | 42.91 (15.15) | 18–100 |
| 18–24 | | 231 (11.5) | | |
| 25–44 | | 922 (45.8) | | |
| 45–64 | | 657 (32.6) | | |
| >= 65 | | 203 (10.1) | | |
| Gender | 2005 | | | |
| Female | | 1685 (84.0) | | |
| Male | | 294 (14.7) | | |
| Other | | 26 (1.3) | | |
| Education | 1991 | | | |
| Elementary | | 4 (0.2) | | |
| Middle school | | 15 (0.8) | | |
| Highschool or equivalent | | 212 (10.6) | | |
| Apprenticeship/trade school | | 31 (1.6) | | |
| College (non-univ) | | 167 (8.4) | | |
| Univ, below bachelor level | | 159 (8.0) | | |
| Univ, bachelor level or higher | | 1403 (70.5) | | |
| Socio-economic status (ladder) | 1952 | | 6.38 (1.70) | 1–10 |
| Country of residence | 1963 | | | |
| North America, Canada and United States | | 762 (38.8) | | |
| European, European Union [EU] members | | 804 (39.9) | | |
| European, non-EU members | | 364 (18.1) | | |
| Other | | 33 (1.6) | | |
| Pre-existing health conditions, Yes (any) | 1943 | 600 (30.9) | | |
| Heart condition or cardiovascular disease | | 68 (3.5) | | |
| Chronic respiratory diseases | | 236 (12.2) | | |
| Type 2 Diabetes | | 49 (2.5) | | |
| Autoimmune disease | | 192 (9.9) | | |
| Currently receiving chemotherapy | | 10 (0.5) | | |
| Other conditions that affect immune function | | 205 (10.7) | | |
| COVID-19 symptoms during past week, Yes (any) | 2012 | 496 (24.7) | | |
| Dry cough | | 218 (10.8) | | |
| Low-grade fever | | 93 (4.6) | | |
| Difficulty breathing | | 94 (4.7) | | |
| Fatigue or muscle pains | | 297 (14.8) | | |
| Tested for COVID-19 | 2010 | | | |
| Yes, test result was positive | | 3 (0.1) | | |
| Yes, test result was negative | | 47 (2.3) | | |
| Yes, don't know the result yet | | 7 (0.3) | | |
| No | | 1953 (97.2) | | |
| Live with someone diagnosed with COVID-19 | 2009 | | | |
| Yes | | 7 (0.3) | | |
| No | | 2002 (99.7) | | |
| Belief in conspiracy theories | 1929 | | 4.2 (2.3) | 1–9 |
| Health literacy | 1934 | | 3.7 (0.6) | 1–4 |
| Prosocial attitudes, Sum score | 1913 | | 6.05 (.97) | 1–7 |
| Distress (PHQ-4), Sum score | 1912 | | 2.10 (.83) | 1–4 |

Note. PHQ-4 –Patient Health Questionnaire-4.

**Table 2.  Endorsement rates motivations "for" and "against" social distancing (N = 2013).**

| Crt no. | Variable | N | % |
|---|---|---|---|
| **Individual-level Motivations** | | | |
| 1 | I want to protect myself | 1690 | 84.0 |
| 2 | I want to avoid spreading the virus to others | 1666 | 82.8 |
| 3 | I am concerned about spreading the virus to vulnerable people. | 1634 | 81.2 |
| 4 | I feel good about myself when I protect others. | 934 | 46.4 |
| 5 | I feel less stressed when I practice social distancing. | 877 | 43.6 |
| 6 | I feel more in control when I practice social distancing | 822 | 40.8 |
| 7 | I don't have a pre-existing medical condition | 611 | 30.4 |
| 8 | I have an elevated risk for COVID-19 | 391 | 19.4 |
| 9 | *I feel stressed when I am alone or in isolation | 268 | 13.3 |
| 10 | *I don't trust the messages my government provides about the pandemic | 255 | 12.7 |
| 11 | *I think it's unlikely I will catch the virus. | 159 | 7.9 |
| 12 | *I cannot afford to pay for delivery for food or groceries | 116 | 5.8 |
| 13 | *I don't like to be told what to do. | 112 | 5.6 |
| 14 | *I believe the best strategy to manage this pandemic is to let the virus run its course. | 78 | 3.9 |
| 15 | *I think the government is exaggerating the impact of this pandemic. | 77 | 3.8 |
| 16 | *I think I cannot spread the virus if I am not sick. | 39 | 1.9 |
| 17 | *I don't have a good internet connection at home | 38 | 1.9 |
| 18 | *I think this pandemic is not serious. | 30 | 1.5 |
| 19 | *I believe prayers and religious rituals can protect me from this virus. | 29 | 1.4 |
| 20 | *I've heard social distancing is not effective at reducing transmission of the virus | 22 | 1.1 |
| **Interpersonal-level Motivations** | | | |
| 21 | I want to protect others. | 1726 | 85.7 |
| 22 | I feel a sense of responsibility to protect our community. | 1688 | 83.9 |
| 23 | I care about the well-being of others. | 1634 | 81.2 |
| 24 | I have friends or family who are vulnerable to the virus. | 1415 | 70.3 |
| 25 | I feel connected to others even when I practice social distancing. | 1135 | 56.4 |
| 26 | I live with someone who is vulnerable to the virus. | 569 | 28.3 |
| 27 | *I have friends or family who need me to run errands for them. | 497 | 24.7 |
| 28 | *I socialize with people to avoid feeling lonely. | 124 | 6.2 |
| 29 | *I don't have friends or family who are vulnerable to the virus. | 84 | 4.2 |
| 30 | *I believe it's OK to invite people to your home to socialize in small groups. | 69 | 3.4 |
| 31 | *I believe my actions cannot protect others from contracting the virus. | 64 | 3.2 |
| 32 | *I believe it's OK to go out and meet with people in small groups. | 59 | 2.9 |
| **Organizational-level Motivations** | | | |
| 33 | My workplace or school recommended we practice social distancing. | 1076 | 53.5 |
| 34 | My workplace or school conducts operations remotely | 1025 | 50.9 |
| 35 | My workplace or school closed down | 712 | 35.4 |
| 36 | My workplace or school discouraged us from coming in | 648 | 32.2 |
| 37 | *My workplace has implemented social distancing policies for workers that have to come to work | 530 | 26.3 |
| 38 | *My work cannot be done remotely. | 324 | 16.1 |
| 39 | *My workplace requires me to come into work. | 224 | 11.1 |
| 40 | *My workplace won't pay me if I do not go into work. | 93 | 4.6 |
| 41 | *My workplace told me that I could lose my job if I do not go into work. | 18 | 0.9 |
| **Community-level Motivations** | | | |
| 42 | Restaurants in my area are closed for eating-in. | 1911 | 94.9 |

(*Continued*)

**Table 2.** (Continued)

| Crt no. | Variable | N | % |
|---|---|---|---|
| 43 | Community centers and recreational facilities in my area are closed. | 1897 | 94.2 |
| 44 | There are no social events held in person in my area. | 1825 | 90.7 |
| 45 | My government says I must do social distancing. | 1777 | 88.3 |
| 46 | My news sources say I should do social distancing. | 1675 | 83.2 |
| 47 | It is possible to shop online and have items delivered to my house. | 1653 | 82.1 |
| 48 | There is barely anyone walking outside in my area. | 1098 | 54.5 |
| 49 | *My place of faith is closed (for example, mosque, temple, church, synagogue). | 916 | 45.5 |
| 50 | *There are many people walking on the streets in my area. | 624 | 31.0 |
| 51 | *It is not possible to shop online and get deliveries in my area. | 215 | 10.7 |
| 52 | *My place of faith is open (for example, mosque, temple, church, synagogue). | 71 | 3.5 |
| 53 | *There are social events held in person in my area. | 36 | 1.8 |
| 54 | *Community centers and recreational facilities in my area are open. | 31 | 1.5 |
| 55 | *Restaurants in my area are open for eating-in | 16 | 0.8 |

Note. Instructions were to check all that apply.

Motivations "for" were conceptualized as facilitators of social distancing.

* Motivations "against' were conceptualized as barriers to social distancing.

## Predictors of adherence to social distancing behaviors

Detailed statistics for regression models predicting social distancing behaviors are included in Table 4.

**Working remotely from home.** Completing a bachelor degree or higher, prosocial attitudes, and motivation for social distancing, i.e., working or attending school remotely, were associated with adherence to working remotely, while "other" gender identity, age of 65 or higher, and barriers to social distancing, i.e., having workplace social distancing measures implemented and unable to do work remotely, were associated with non-adherence.

**Avoiding contact outside of one's household.** Health literacy, prosocial attitudes, and motivation for social distancing, i.e., feeling responsible for protecting the community, were associated with adherence to avoiding contact outside of one's household, while residing in a country with police-enforced social distancing measures, having a pre-existing medical condition, believing in conspiracies, and barriers to social distancing, i.e., having to run errands for friends/family and seeing many people out on the streets, were related to non-adherence.

**Avoiding in-person socializing.** Female gender, 45 years of age or older, distress, and motivations for social distancing, i.e., wanting to protect the self, were associated with adherence to avoiding in-person socializing while barriers to social distancing, i.e., feeling stressed or alone or in isolation, having to run errands for friends/family, socializing to avoid loneliness, and seeing many people out on the streets, were associated with non-adherence.

**Maintaining a 2-m distance from others.** Female gender, being 25 years of age or older, prosocial attitudes, and motivation for social distancing, i.e., wanting to protect others, feeling responsible for the community, were associated with adherence to maintaining a safe distance from others, while barriers to social distancing, i.e., unable to do work remotely, and seeing many people out on the streets, were related to non-adherence.

**Avoiding getting out of the house.** Female gender, prosocial attitudes, and motivation for social distancing, i.e., wanting to protect the self and others, were associated with avoiding leaving the home while living in a country with police-enforced social distancing measures,

**Table 3. Descriptive statistics for adherence to social distancing recommendations (N = 2,013).**

| Crt no | Variable | M (SD) | Never (%) | Sometimes (%) | Often (%) | Always (%) | N/A (%) | Always (%) Week 1 | Week 2 | Week 3 | p |
|---|---|---|---|---|---|---|---|---|---|---|---|
| 1 | Working from home or remotely | 3.2 (1.2) | 12.0 | 9.3 | 9.4 | 44.8 | 24.5 | 46.8 | 42.3 | 46.7 | - |
| 2 | Attending classes virtually or completing coursework remotely | 2.7 (1.2) | 13.1 | 9.9 | 9.2 | 20.6 | 47.2 | 19.4 | 19.6 | 24.3 | < .001 |
| 3 | Avoiding non-essential gatherings (social events) | 3.9 (0.5) | 2.1 | 1.5 | 4.9 | 88.7 | 2.7 | 89.6 | 88.3 | 88.3 | - |
| 4 | Avoiding crowded places (concerts, conferences, arenas, festivals) | 3.9 (0.4) | 0.9 | 0.7 | 2.0 | 90.6 | 5.8 | 92.4 | 88.8 | 91.6 | .03 |
| 5 | Avoiding going out to restaurants, bars, pubs, coffee shops, etc. | 3.9 (0.4) | 1.0 | 1.1 | 3.6 | 88.8 | 5.5 | 90.6 | 86.4 | 91.0 | .001 |
| 6 | Avoiding any non-essential travel (domestic, international) | 3.9 (0.4) | 1.0 | 0.9 | 3.8 | 90.5 | 3.7 | 92.3 | 87.9 | 92.9 | - |
| 7 | Avoiding common greetings that involve close contact (hugs, kisses, handshakes) | 3.9 (0.5) | 1.3 | 1.7 | 6.4 | 88.1 | 2.5 | 90.2 | 87.2 | 86.8 | - |
| 8 | Avoiding making contact with family members who do not typically live with you | 3.7 (0.7) | 2.8 | 4.5 | 14.2 | 73.6 | 5.0 | 73.9 | 73.2 | 73.8 | - |
| 9 | Avoiding socializing in person even with close friends | 3.8 (0.6) | 1.6 | 3.0 | 12.3 | 82.0 | 1.0 | 81.6 | 81.8 | 83.1 | - |
| 10 | Avoiding or limiting contact with people at higher risk or vulnerable populations (for example, older adults, those with at risk conditions or those in poor health) | 3.9 (0.5) | 1.4 | 1.8 | 7.0 | 85.0 | 4.8 | 88.5 | 84.2 | 86.0 | - |
| 11 | Ordering take-out from restaurants (picked up in person) | 1.7 (1.0) | 54.1 | 23.1 | 5.6 | 8.2 | 9.0 | 9.4 | 7.6 | 7.9 | - |
| 12 | Having meals/groceries delivered to your house | 1.8 (.1.0) | 49.7 | 27.3 | 8.8 | 8.7 | 5.4 | 8.5 | 8.9 | 8.8 | - |
| 13 | Keeping a safe distance of at least 6 feet (approximately 2 meters) | 3.6 (0.6) | 0.5 | 3.5 | 28.4 | 66.2 | 1.4 | 66.1 | 66.6 | 65.5 | .02 |
| 14 | Isolating myself at home, when sick | 3.8 (0.6) | 1.4 | 1.2 | 2.8 | 45.9 | 48.7 | 45.7 | 43.3 | 50.8 | .001 |
| 15 | Avoiding leaving the home, except to go to grocery store or pharmacy | 3.6 (0.7) | 2.3 | 7.7 | 20.5 | 66.9 | 2.6 | 69.3 | 70.2 | 57.3 | < .001 |

Note. Response options for each item ranged from 1 ('Never') to 4 ('Always').

having a pre-existing medical condition, and barriers for social distancing, i.e., feeling stresses when alone/in isolation, and seeing many people out on the streets, were related to non-adherence.

## Discussion

The current study investigated rates of motivations (or barriers and facilitators) for social distancing and adherence to social distancing recommendations in a large convenience sample of 2013 English-speaking adults recruited primarily from Europe and North America. Data were collected during a period of time where across most countries in Europe and North America regulations about social distancing were relatively strict (e.g., shelter-in-place and working from home orders). Our results suggest that individuals are motivated to engage in social distancing by both internal factors, including wanting to protect self and others, wanting to avoid spreading the virus to others and feeling the responsibility to protect the community, as well as external circumstances, including institutions conducting work remotely and social events being cancelled. Prioritizing one's health has also been reported as motivating factors in

**Table 4. Stepwise logistic regression models predicting adherence to social distancing behaviors.**

| Variable | Working remotely | | | Avoiding contact outside of household | | | Avoiding socializing in person even with close friends | | | Keeping a safe distance of at least 6 feet (approximately 2 meters) | | | Avoiding leaving the home, except for grocery store or pharmacy | | |
|---|---|---|---|---|---|---|---|---|---|---|---|---|---|---|---|
| | Adherent | Non adherent | Adj OR | Adherent | Non adherent | Adj OR | Adherent | Non adherent | Adj OR | Adherent | Non adherent | Adj OR | Adherent | Non adherent | Adj OR |
| Gender, n (%) | | | | | | | | | | | | | | | |
| Male | 138 | 88 | REF | 205 | 64 | REF | 206 | 85 | REF | 170 | 120 | REF | 170 | 119 | REF |
| | (61.1) | (38.9) | | (76.2) | (23.8) | | (70.8) | (29.2) | | (58.6) | (41.4) | | (58.8) | (41.2) | |
| Female | 753 | 515 | 0.84 | 1254 | 360 | 0.95 | 1420 | 250 | **2.02** | 1146 | 516 | **1.50** | 1159 | 482 | **1.42** |
| | (59.4) | (40.6) | [0.57, 1.25] | (77.7) | (22.3) | [0.67, 1.34] | (85.0) | (15.0) | **[1.45, 2.82]** | (69.0) | (31.0) | **[1.11, 2.03]** | (70.6) | (29.4) | **[1.05, 1.91]** |
| Other | 6 | 13 | **0.21** | 18 | 5 | 1.27 | 20 | 4.0 | 2.79 | 13 | 12 | 1.12 | 13 | 11 | 0.85 |
| | (31.6) | (68.4) | **[0.06, 0.78]** | (78.3) | (21.7) | [0.42, 3.79] | (83.3) | (16.7) | [0.87, 8.98] | (52.0) | (48.0) | [0.45, 2.76] | (54.2) | (45.8) | [0.33, 2.16] |
| Age, n (%) | | | | | | | | | | | | | | | |
| 18–24 | 88 | 58 | REF | 157 | 65 | REF | 158 | 72 | REF | 109 | 114 | REF | 137 | 90 | REF |
| | (60.3) | (39.7) | | (70.7) | (29.3) | | (68.7) | (31.3) | | (48.9) | (51.1) | | (60.4) | (39.6) | |
| 25–44 | 507 | 294 | 0.98 | 683 | 193 | 1.21 | 751 | 165 | 1.44 | 539 | 376 | **1.45** | 603 | 306 | 1.09 |
| | (63.3) | (36.7) | [0.58, 1.63] | (78.0) | (22.0) | [0.81, 1.80] | (82.0) | (18.0) | [0.96, 2.15] | (58.9) | (41.1) | **[1.02, 2.07]** | (66.3) | (33.7) | [0.76, 1.58] |
| 45–64 | 283 | 222 | 0.96 | 486 | 143 | 1.21 | 561 | 84 | **2.27** | 510 | 140 | **3.25** | 462 | 167 | 1.24 |
| | (56.0) | (44.0) | [0.56, 1.66] | (77.3) | (22.7) | [0.79, 1.85] | (87.0) | (13.0) | **[1.43, 3.59]** | (78.5) | (21.5) | **[2.20, 4.81]** | (73.4) | (26.6) | [0.83, 1.86] |
| > = 65 | 23 | 45 | **0.39** | 155 | 30 | 1.65 | 181 | 20 | **2.55** | 174 | 22 | **6.77** | 144 | 52 | 1.06 |
| | (33.8) | (66.2) | **[0.18, 0.88]** | (83.8) | (16.2) | [0.90, 3.01] | (90.0) | (10.0) | **[1.32, 4.95]** | (88.8) | (11.2) | **[3.65, 12.57]** | (73.5) | (26.5) | [0.63, 1.79] |
| Education, n (%) | | | | | | | | | | | | | | | |
| < Bachelor | 164 | 194 | REF | 414 | 144 | REF | 466 | 113 | REF | 397 | 178 | REF | 398 | 168 | REF |
| | (45.8) | (54.2) | | (74.2) | (25.8) | | (80.5) | (19.5) | | (69.0) | (31.0) | | (70.3) | (29.7) | |
| > = Bachelor | 731 | 417 | **1.48** | 1054 | 279 | 1.25 | 1168 | 224 | 1.14 | 919 | 469 | 0.77 | 932 | 442 | 0.90 |
| | (63.7) | (36.3) | **[1.06, 2.08]** | (79.1) | (20.9) | [0.95, 1.65] | (83.9) | (16.1) | [0.84, 1.56] | (66.2) | (33.8) | [0.59, 1.00] | (67.8) | (32.2) | [0.70, 1.17] |
| SES-ladder | 6.5 ± 1.6 | 6.3 ± 1.8 | 0.97 | 6.4 ± 1.7 | 6.3 ± 1.7 | 1.00 | 6.4 ± 1.7 | 6.4 ± 1.8 | 0.95 | 6.5 ± 1.7 | 6.3 ± 1.7 | 1.03 | 6.3 ± 1.7 | 6.5 ± 1.7 | 0.96 |
| | | | [0.90, 1.06] | | | [0.93, 1.08] | | | [0.87, 1.03] | | | [0.96, 1.10] | | | [0.90, 1.03] |
| **Country of residence | | | | | | | | | | | | | | | |
| +Moderate rules | 515 | 349 | REF | 848 | 255 | REF | 949 | 200 | REF | 783 | 359 | REF | 709 | 416 | REF |
| | (59.6) | (40.4) | | (76.9) | (23.1) | | (82.6) | (17.4) | | (68.6) | (31.4) | | (63.0) | (37.0) | |
| +Strict rules | 356 | 246 | 1.06 | 589 | 154 | **0.73** | 651 | 122 | 0.90 | 504 | 268 | 1.15 | 594 | 173 | **0.46** |
| | (59.1) | (40.9) | [0.79, 1.41] | (79.3) | (20.7) | **[0.57, 0.95]** | (84.2) | (15.8) | [0.68, 1.20] | (65.3) | (34.7) | [0.91, 1.44] | (77.4) | (22.6) | **[0.36, 0.59]** |
| Pre-existing conditions, n (%) | | | | | | | | | | | | | | | |
| No | 628 | 415 | REF | 975 | 300 | REF | 1090 | 242 | REF | 859 | 467 | REF | 853 | 455 | REF |
| | (60.2) | (39.8) | | (70.7) | (29.3) | | (81.8) | (18.2) | | (64.8) | (35.2) | | (65.2) | (34.8) | |
| Yes | 245 | 178 | 1.02 | 464 | 109 | **0.71** | 509 | 83 | 0.81 | 424 | 166 | 0.81 | 446 | 141 | **0.66** |
| | (57.9) | (42.1) | [0.75, 1.39] | (70.7) | (29.3) | **[0.54, 0.95]** | (86.0) | (14.0) | [0.59, 1.11] | (71.9) | (28.1) | [0.63, 1.04] | (76.0) | (24.0) | **[0.51, 0.86]** |
| COVID-19 symptoms, n (%) | | | | | | | | | | | | | | | |
| No | 695 | 457 | REF | 1117 | 320 | REF | 1249 | 251 | REF | 1018 | 475 | REF | 1001 | 482 | REF |
| | (60.3) | (39.7) | | (70.7) | (29.3) | | (83.3) | (16.7) | | (68.2) | (31.8) | | (67.5) | (32.5) | |
| Yes | 206 | 161 | 0.96 | 364 | 110 | 1.09 | 402.0 | 89.0 | 1.14 | 314 | 176 | 1.10 | 345 | 132 | 0.77 |
| | (56.1) | (43.9) | [0.70, 1.32] | (70.7) | (29.3) | [0.82, 1.44] | (81.9) | (18.1) | [0.84, 1.55] | (64.1) | (35.9) | [0.85, 1.41] | (72.3) | (27.7) | [0.59, 1.00] |
| **Psychological variables, M ± SD** | | | | | | | | | | | | | | | |

(*Continued*)

**Table 4.** (*Continued*)

| Variable | Working remotely | | | Avoiding contact outside of household | | | Avoiding socializing in person even with close friends | | | Keeping a safe distance of at least 6 feet (approximately 2 meters) | | | Avoiding leaving the home, except for grocery store or pharmacy | | |
|---|---|---|---|---|---|---|---|---|---|---|---|---|---|---|---|
| | Adherent | Non adherent | Adj OR | Adherent | Non adherent | Adj OR | Adherent | Non adherent | Adj OR | Adherent | Non adherent | Adj OR | Adherent | Non adherent | Adj OR |
| Conspiracy beliefs | 4.0 ± 2.3 | 4.3 ± 2.3 | 0.95 [0.89, 1.01] | 4.1 ± 2.3 | 4.5 ± 2.3 | **0.93** **[0.88, 0.98]** | 4.2 ± 2.3 | 4.3 ± 2.3 | 0.99 [0.93, 1.05] | 4.1 ± 2.3 | 4.3 ± 2.2 | 0.98 [0.94, 1.03] | 4.2 ± 2.3 | 4.1 ± 2.3 | 1.00 [0.95, 1.05] |
| Health literacy | 3.7 ± 0.6 | 3.6 ± 0.6 | 1.25 [0.99, 1.58] | 3.7 ± 0.6 | 3.6 ± 0.7 | **1.43** **[1.18, 1.72]** | 3.7 ± 0.6 | 3.6 ± 0.6 | 1.19 [0.96, 1.48] | 3.7 ± 0.6 | 3.6 ± 0.6 | 1.14 [0.95, 1.37] | 3.7 ± 0.6 | 3.7 ± 0.6 | 1.18 [0.98, 1.42] |
| PhQ-4 | 2.1 ± 0.8 | 2.1 ± 0.8 | 1.15 [0.96, 1.38] | 2.1 ± 0.8 | 2.1 ± 0.8 | 1.12 [0.96, 1.32] | 2.1 ± 0.8 | 2.1 ± 0.8 | **1.31** **[1.09, 1.57]** | 2.1 ± 0.8 | 2.2 ± 0.8 | 1.08 [0.93, 1.24] | 2.1 ± 0.8 | 2.1 ± 0.8 | 1.09 [0.94, 1.26] |
| Prosocial attitudes | 6.1 ± 1.0 | 6.0 ± 1.0 | **1.17** **[1.02, 1.35]** | 6.1 ± 1.0 | 6.0 ± 0.9 | **1.13** **[1.00, 1.28]** | 6.1 ± 1.0 | 5.9 ± 1.0 | 1.10 [0.96, 1.26] | 6.2 ± 0.9 | 5.9 ± 1.0 | **1.22** **[1.09, 1.37]** | 6.1 ± 1.0 | 5.9 ± 0.9 | **1.16** **[1.04, 1.30]** |
| **Individual-level motivations (yes), n (%)** | | | | | | | | | | | | | | | |
| Want to protect myself | 754 (59.5) | 514 (40.5) | 1.04 [0.70, 1.55] | 1270 (78.8) | 342 (21.2) | 1.37 [0.98, 1.90] | 1427 (85.0) | 252 (15.0) | **2.17** **[1.54, 3.05]** | 1167 (69.8) | 506 (30.2) | **1.62** **[1.20, 2.19]** | 1175 (71.2) | 475 (28.8) | **1.93** **[1.43, 2.61]** |
| Want to avoid spreading virus | 744 (59.2) | 512 (40.8) | 0.84 [0.55, 1.29] | 1240 (78.2) | 346 (21.8) | 0.98 [0.68, 1.42] | 1394 (84.2) | 262 (15.8) | 1.41 [0.96, 2.08] | 1132 (68.8) | 513 (31.2) | 1.02 [0.73, 1.42] | 1133 (69.8) | 490 (30.2) | 0.99 [0.71, 1.39] |
| *Don't have a pre-existing medical condition | 280 (60.7) | 181 (39.3) | 1.11 [0.82, 1.51] | 447 (76.8) | 135 (23.2) | 1.05 [0.80, 1.37] | 487 (80.6) | 117 (19.4) | 1.08 [0.80, 1.45] | 391 (64.6) | 214 (35.4) | 0.96 [0.75, 1.22] | 378 (63.3) | 219 (36.7) | 0.81 [0.64, 1.03] |
| *Feel stressed when I'm alone or in isolation | 118 (58.1) | 85 (41.9) | 0.68 [0.44, 1.04] | 185 (72.5) | 70 (27.5) | 0.94 [0.66, 1.35] | 195 (73.3) | 71 (26.7) | **0.56** **[0.38, 0.81]** | 153 (58.4) | 109 (41.6) | 0.97 [0.70, 1.34] | 157 (59.0) | 109 (41.0) | **0.71** **[0.51, 0.98]** |
| **Interpersonal-level motivations (yes), n (%)** | | | | | | | | | | | | | | | |
| Want to protect others | 775 (59.5) | 528 (40.5) | 1.03 [0.64, 1.66] | 1285 (78.1) | 360 (21.9) | 0.95 [0.64, 1.42] | 1430 (83.5) | 282 (16.5) | 0.75 [0.48, 1.17] | 1181 (69.2) | 526 (30.8) | 1.35 [0.94, 1.95] | 1187 (70.6) | 495 (29.4) | **1.53** **[1.07, 2.20]** |
| Feel a sense of responsibility to protect our community | 769 (60.1) | 510 (39.9) | 1.18 [0.78, 1.79] | 1273 (79.1) | 337 (20.9) | **1.52** **[1.08, 2.15]** | 1408 (84.0) | 268 (16.0) | 1.22 [0.83, 1.78] | 1164 (69.9) | 502 (30.1) | **1.47** **[1.07, 2.02]** | 1159 (70.5) | 486 (29.5) | 1.25 [0.90, 1.74] |
| *Have friends/family who need me to run errands | 217 (57.3) | 162 (42.7) | 0.79 [0.58, 1.09] | 333 (70) | 143 (30.0) | **0.54** **[0.41, 0.70]** | 393 (80.2) | 97 (19.8) | **0.62** **[0.46, 0.84]** | 334 (67.5) | 161 (32.5) | 0.99 [0.77, 1.27] | 327 (67.0) | 161 (33.0) | 0.80 [0.63, 1.03] |
| *I socialize with people to avoid feeling lonely | 70 (68.0) | 33 (32.0) | 1.43 [0.82, 2.50] | 76 (65.5) | 40 (34.5) | 0.68 [0.44, 1.07] | 81 (65.3) | 43 (34.7) | **0.51** **[0.33, 0.79]** | 61 (49.6) | 62 (50.4) | 0.69 [0.45, 1.04] | 65 (52.4) | 59 (47.6) | 0.70 [0.46, 1.06] |
| **Organizational-level motivations (yes), n (%)** | | | | | | | | | | | | | | | |
| My work/school recommended we practice social distancing | 607 (64.4) | 336 (35.6) | 1.29 [0.94, 1.77] | 803 (78.0) | 227 (22.0) | 1.25 [0.94, 1.67] | 874 (81.7) | 196 (18.3) | 0.96 [0.70, 1.33] | 672 (63.0) | 394 (37.0) | 0.86 [0.66, 1.11] | 705 (67.3) | 343 (32.7) | 1.15 [0.88, 1.50] |
| My work/school conducts operations remotely | 719 (77.0) | 215 (23.0) | **4.66** **[3.46, 6.26]** | 752 (76.5) | 231 (23.5) | 0.82 [0.61, 1.08] | 834 (82.0) | 183 (18.0) | 1.05 [0.76, 1.43] | 638 (62.9) | 376 (37.1) | 0.86 [0.67, 1.11] | 674 (66.5) | 340 (33.5) | 0.83 [0.64, 1.07] |
| *My work has implemented SD policies for workers | 261 (53.5) | 227 (46.5) | **0.61** **[0.44, 0.83]** | 389 (76.9) | 117 (23.1) | 0.94 [0.70, 1.27] | 434 (82.5) | 92 (17.5) | 0.96 [0.70, 1.33] | 345 (65.7) | 180 (34.3) | 1.02 [0.78, 1.33] | 342 (66.8) | 170 (33.2) | 0.96 [0.73, 1.25] |
| *My work cannot be done remotely | 24 (9.3) | 234 (90.7) | **0.07** **[0.04, 0.11]** | 232 (74.8) | 78 (25.2) | 0.82 [0.58, 1.16] | 258 (80.6) | 62 (19.4) | 0.88 [0.61, 1.29] | 198 (61.3) | 125 (38.7) | **0.69** **[0.50, 0.93]** | 186 (62.2) | 113 (37.8) | 0.71 [0.51, 0.97] |

(*Continued*)

**Table 4.** (Continued)

| Variable | Working remotely | | | Avoiding contact outside of household | | | Avoiding socializing in person even with close friends | | | Keeping a safe distance of at least 6 feet (approximately 2 meters) | | | Avoiding leaving the home, except for grocery store or pharmacy | | |
|---|---|---|---|---|---|---|---|---|---|---|---|---|---|---|---|
| | Adherent | Non adherent | Adj OR | Adherent | Non adherent | Adj OR | Adherent | Non adherent | Adj OR | Adherent | Non adherent | Adj OR | Adherent | Non adherent | Adj OR |
| **Community-level motivations (yes), n (%)** | | | | | | | | | | | | | | | |
| Recreational facilities closed | 853 | 577 | 1.29 | 1402 | 403 | 1.00 | 1563 | 320 | 0.86 | 1261 | 612 | 1.17 | 1270 | 583 | 0.93 |
| | (59.7) | (40.3) | [0.65, 2.58] | (77.7) | (22.3) | [0.55, 1.83] | (83.0) | (17.0) | [0.43, 1.75] | (67.3) | (32.7) | [0.68, 2.03] | (68.5) | (31.5) | [0.52, 1.67] |
| Restaurants closed for eating in | 858 | 580 | 0.72 | 1415 | 405 | 1.32 | 1577 | 321 | 1.15 | 1270 | 617 | 1.21 | 1277 | 588 | 0.83 |
| | (59.7) | (40.3) | [0.34, 1.52] | (77.7) | (22.3) | [0.70, 2.49] | (83.1) | (16.9) | [0.56, 2.39] | (67.3) | (32.7) | [0.67, 2.19] | (68.5) | (31.5) | [0.44, 1.57] |
| *There are many people walking on the streets | 305 | 197 | 1.25 | 444 | 156 | **0.76** | 486 | 135 | **0.69** | 360 | 260 | **0.66** | 356 | 255 | **0.57** |
| | (60.8) | (39.2) | [0.93, 1.68] | (74.0) | (26.0) | **[0.59, 0.98]** | (78.3) | (21.7) | **[0.52, 0.91]** | (58.1) | (41.9) | **[0.52, 0.82]** | (58.3) | (41.7) | **[0.45, 0.71]** |
| *Not possible to shop online and get deliveries | 81 | 63 | 1.19 | 158 | 39 | 1.42 | 182 | 29 | 1.39 | 144 | 66 | 0.78 | 160 | 51 | 1.29 |
| | (56.3) | (43.8) | [0.75, 1.90] | (80.2) | (19.8) | [0.92, 2.17] | (86.3) | (13.7) | [0.85, 2.26] | (68.6) | (31.4) | [0.54, 1.12] | (75.8) | (24.2) | [0.88, 1.89] |

Note.

*Motivations "against" social distancing. PHQ-4 = Patient Health Questionnaire-4. SES = socio-economic status.

**Countries in North America and Europe with moderate or strict regulations about social distancing.

+Countries with moderate social distancing restrictions included: Canada, the United States, Belgium, Croatia, Estonia, Germany, Latvia, Lichtenstein, Lithuania, Luxembourg, Malta, Poland, Slovakia, Slovenia, Switzerland, the United Kingdom.

+Countries with strict social distancing regulations, such as police enforced isolation, included Albania, Andorra, Austria, Bosnia and Herzegovina, Bulgaria, Czech Republic, France, Greece, Hungary, Ireland, Italy, Monaco, Montenegro, Portugal, Republic of Moldova, Romania, Russian Federation, Serbia, Spain, The Former Yugoslav Republic, Ukraine.

Participants from European countries with minimal to no regulations about social distancing, including Denmark (n = 1), Finland (n = 1), Iceland (n = 1), Netherlands (n = 11), and Sweden (n = 1) were excluded from analyses.

another large international survey, in addition to believing that adhering to social distancing behaviours will be effective in preventing COVID-19 [45]. Key barriers against social distancing included feeling stressed when alone, socializing to avoid loneliness, having to run errands for family or friends, not being able to do work remotely, and seeing many people on the streets in the area of residence. Importantly, barriers tapping misconceptions and/or conspiratory beliefs, such as inability to pass the virus unless sick or showing symptoms, the government exaggerating the impact of the epidemic, social distancing not being effective at reducing virus transmission, or "letting the virus run its course", were endorsed only by a small minority of respondents (1–3%).

Adherence to social distancing behaviours that are within one's control, such as avoiding non-essential travel, social gatherings, or handshakes, was relatively high, yet not perfect. This is consistent with preliminary results from the international iCARE study, which also showed high self-reported adherence to avoiding gatherings, staying 1–2 metres away from others, staying home and avoiding the grocery store [46]. While the rates of adherence in the current study are very promising, it is likely closely linked to the timeline for data collection which paralleled strictly enforced social distancing regulations worldwide. Notably, adherence to behaviours taxed by external circumstances (e.g., institutional policies or geographic location) including working remotely or ordering meals online, was lower. These findings have direct implications to future public health recommendations during this pandemic, especially as several countries are moving towards relaxing social distancing measures currently in place. It is

possible that as distancing measures relax, adherence rates to social distancing recommendations would decrease, potentially causing a spike in COVID-19 incidence once again [13]. In fact, given that 25% of our sample reported symptoms consistent with COVID-19 but only 3% had been tested, it is imperative that public health initiatives focus on wider scale testing and contact tracing coupled with continued recommendations for social distancing and proper hygiene.

Based on logistic regression models which examined the impact of socio-demographic, medical, and psychological predictors on adherence to social distancing recommendations, we found that women were more likely than men and older individuals (> 45 years old) were more likely than younger individuals (18–24 years old) to avoid socializing in person and maintain a safe distance when in public. Wanting to protect the self, feeling a responsibility to protect the community, and being able to work or study remotely were the strongest predictors for adherence to social distancing recommendations. In contrast, individuals living in a country with more strict rules for social distancing and those having at least one pre-existing medical condition were less likely to avoid meeting with family members outside of the household and to leave home only for grocery or pharmacy trips. Similarly, having friends or family who needed help with running errands, socializing in order to avoid feeling lonely, and seeing many people in the streets were the strongest barriers to adherence to social distancing recommendations.

## Limitations

There are limitations to this study, including the use of a convenience sample (i.e., recruited on social media) and homogeneity of sample characteristics (i.e., 84% female, >70% completed at least a bachelor's degree), which might affect the generalizability of our findings to predominantly male samples, more diverse samples, or individuals without easy access to the internet and social media platforms (Facebook, Twitter). The cross-sectional design allowed for testing of associations between predictor and outcome variables at one point in time, but longitudinal predictions could not be made. Lastly, the outcome variables, social distancing behaviours, and some of the predictor variables, including motivations for social distancing, were created by the authors specifically for this study and their psychometric properties are currently unknown. As such, it is possible our measures did not cover an exclusive range of behaviors and/or motivations about social distancing. In hindsight, we believe it would have been beneficial to ask respondents to identify the levels of restrictions in their area of residence, as opposed to inferring the restrictions based on their country of residence.

## Implications and future directions

Results from the current study suggest that men are less adherent (~30–40% non-adherence) to social distancing recommendations compared to women (~15–30% non-adherence). These findings are consistent with another international survey which found women were more compliant to sheltering-in-place rules [45]. This finding may be explained by gender-specific differences in health information speaking behaviour, where women are more likely to actively seek out and engage with health information [47] and risk tolerance and behaviour which is generally lower in women [48]. It should be noted that substantially fewer men compared to women participated in this study, so differences relating to gender might be a function of our unbalanced enrollment. Similarly, younger individuals, particularly in the 18–24 age group, who were found to be less likely than older individuals to adhere to social distancing recommendations, might have a stronger preference or need to socialize in person to seek and receive social support and facilitate relatedness or feelings of belongingness [25, 49]. Public

health communications about social distancing should incorporate more nuanced guidelines for safely engaging socially with others, while also maintaining appropriate physical distancing standards, using non-blaming and non-stigmatizing language and targeting specific groups such as men and younger individuals. In line with our findings that wanting to protect self, others and the community were the strongest motivators associated with higher adherence to social distancing recommendations, it is important that these compassionate-focused and pro-social attitudes are kept at the center of future public health campaigns about social distancing [20–22]. Lastly, the study found that seeing many people walking in the streets was one of the strongest barriers against social distancing (i.e., associated with lower adherence), but seeing few people in the streets acted as a facilitator of adherence to social distancing (statistic not reported). This highlights the importance of social norms and their impact on the uptake of protective behaviors [50].

Future interventions should target modifiable barriers to social distancing identified herein using strategies designed specifically to improve individual motivation to initiate health protective behaviors (e.g., motivational interviewing), tap personal values around self-protection and protecting of others (e.g., acceptance and commitment therapy) and reduce peer pressure to socialize freely with others for those witnessing crowds or living in crowded areas (e.g., compassion-focused therapy). These individual-level interventions coupled with effective organizational measures and community-based or public health interventions will be extremely important to facilitate the uptake and maintenance of social distancing behaviours among the general population until an effective vaccine and /or treatment for COVID-19 is discovered. Future research could investigate the relationship between motivations and adherence to social distancing recommendations in a longitudinal design, which will be relevant especially as countries transition to "opening up" scenarios where restrictions on social interactions will relax. Future research is also needed to establish the psychometric properties of the measures developed specifically for this study, including the assessment or social distancing behaviours and the motivations for social distancing.

## Conclusion

This cross-sectional study collected data from 2013 participants recruited via social media. The study was conducted during a period of well-enforced regulations about social distancing. Adherence to social distancing recommendations was relatively high for most behaviours, but not nearly close to 100%. The study identified key modifiable barriers and facilitators of adherence to social distancing: strongest facilitators included wanting to protect the self, feeling a responsibility to protect the community, and being able to work/study remotely; strongest barriers included having friends or family who needed help with running errands, socializing in order to avoid feeling lonely, and seeing many people in the streets. Future interventions to improve adherence to social distancing measures should couple individual-level strategies targeting key barriers to social distancing identified herein, with effective institutional measures and public health interventions. Public health campaigns should continue to highlight compassionate attitudes towards social distancing.

## Acknowledgments

We would like to thank Gerald Jordan, PhD and Kyla Brophy, MA, MSc for their support with selecting the study measures. We would also like to thank our participants for taking the time to fill out our survey during the COVID-19 pandemic.

## Author Contributions

**Conceptualization:** Adina Coroiu, Chelsea Moran.

**Data curation:** Adina Coroiu, Chelsea Moran.

**Formal analysis:** Adina Coroiu, Chelsea Moran.

**Investigation:** Adina Coroiu, Chelsea Moran, Tavis Campbell, Alan C. Geller.

**Methodology:** Adina Coroiu, Chelsea Moran, Alan C. Geller.

**Project administration:** Adina Coroiu, Chelsea Moran.

**Supervision:** Alan C. Geller.

**Writing – original draft:** Adina Coroiu.

**Writing – review & editing:** Adina Coroiu, Chelsea Moran, Tavis Campbell, Alan C. Geller.

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
