## [Decision Letter · Decision Letter 0]

29 Jun 2020

PONE-D-20-13845

Barriers and Facilitators of Adherence to Social Distancing Recommendations among a Large International Sample of Adults Recruited in April 2020

PLOS ONE

Dear Dr. Coroiu,

Thank you for submitting your manuscript to PLOS ONE. After careful consideration, we feel that it has merit but does not fully meet PLOS ONE’s publication criteria as it currently stands. Therefore, we invite you to submit a revised version of the manuscript that addresses the points raised during the review process.

Please find below the reviewer's and mine's comments.

We look forward to receiving your revised manuscript.

Kind regards,

Valerio Capraro

Academic Editor

PLOS ONE

Journal Requirements:

2. Please include additional information regarding the survey or questionnaire used in the study and ensure that you have provided sufficient details that others could replicate the analyses.

For instance, if you developed a questionnaire as part of this study and it is not under a copyright more restrictive than CC-BY, please include a copy, in both the original language and English, as Supporting Information.

3. Your ethics statement must appear in the Methods section of your manuscript. If your ethics statement is written in any section besides the Methods, please move it to the Methods section and delete it from any other section.

Please also ensure that your ethics statement is included in your manuscript, as the ethics section of your online submission will not be published alongside your manuscript.

4.Thank you for stating the following in the Title Page of your manuscript:

'This is an unfunded study. AC is supported by post-doctoral research fellowships from the Canadian Institutes

of Health Research (CIHR) and Fonds de Recherche du Quebec – Santé (FRQS). CM is supported by a Vanier

Canada Graduate Scholarship and a University of Calgary Training in Research and Clinical Trials in Integrative

Oncology (TRACTION) fellowship.'

 'The author(s) received no specific funding for this work.'

Additional Editor Comments:

I have now collected one review from one expert in the field. I was unable to find a second reviewer. However, the review I could collect is very detailed, and I am myself familiar with the emerging literature on Covid-19, therefore I feel comfortable in making a decision with only one review. The review is positive and suggests major revision. I agree with the reviewer and therefore I would like to invite you to revise your work for Plos One. Apart from the reviewer's comments, I would like to suggest another improvement. While reading the manuscript, I had the feeling that the emerging literature on Covid-19 was largely neglected. This point should be improved. A good starting point is Van Bavel et al.'s "perspective article" on what social and behavioral science can do to promote pandemic response, published in Nature Human Behaviour. Also, there have been several works testing which appeals and messages promote pandemic response (Bilancini et al. 2020; Capraro & Barcelo, 2020a; Capraro & Barcelo, 2020b; Everett et al. 2020; Heffner et al. 2020; Jordan et al. 2020). Of course it is not a requirement to cite exactly these works, but in any case I think that you should do a much better job at placing your work within the current academic discussion.

Looking forward for the revision.

References

Bilancini E, Boncinelli L, Capraro V, Celadin T, Di Paolo R (2020) The effect of norm-based messages on reading and understanding COVID-19 pandemic response governmental rules. Journal of Behavioral Economics for Policy 4, 45-55.

Capraro, V., & Barcelo, H. (2020a). The effect of messaging and gender on intentions to wear a face covering to slow down COVID-19 transmission. arXiv preprint arXiv:2005.05467.

Capraro, V., & Barcelo, H. (2020b). Priming reasoning increases intentions to wear a face covering to slow down COVID-19 transmission. arXiv preprint arXiv:2006.11273.

Everett, J. A., Colombatto, C., Chituc, V., Brady, W. J., & Crockett, M. (2020). The effectiveness of moral messages on public health behavioral intentions during the COVID-19 pandemic. https://psyarxiv.com/9yqs8/

Heffner, J., Vives, M. L., & FeldmanHall, O. (2020). Emotional responses to prosocial messages increase willingness to self-isolate during the COVID-19 pandemic. https://psyarxiv.com/qkxvb/download?format=pdf

Jordan, J., Yoeli, E., & Rand, D. (2020). Don’t get it or don’t spread it? Comparing self-interested versus prosocially framed COVID-19 prevention messaging. https://psyarxiv.com/yuq7x

Van Bavel, J. J., et al. (2020). Using social and behavioural science to support COVID-19 pandemic response. Nature Human Behaviour.

Reviewers' comments:

Reviewer's Responses to Questions

**Comments to the Author**

1. Is the manuscript technically sound, and do the data support the conclusions?

Reviewer #1: Yes

2. Has the statistical analysis been performed appropriately and rigorously? 

Reviewer #1: Yes

3. Have the authors made all data underlying the findings in their manuscript fully available?

Reviewer #1: Yes

4. Is the manuscript presented in an intelligible fashion and written in standard English?

Reviewer #1: Yes

5. Review Comments to the Author

Reviewer #1: Comments – PONE-D-20-13845

This paper describes the most frequently endorsed motivations to engage in social distancing and the rates of adherence to social distancing recommendations and examines the predictors of adherence to social distancing recommendations, using cross-sectional data collected online on more than 2,000 English-speaking adults from Europe and North America. The topic is undoubtedly relevant, and the findings are very important. However, the statistical approach can be improved, and the results section is difficult to read making the results hard to digest at first. I have a number of questions and suggestions for edits/adjustments. Below are some major and minor points.

Major points

Background

The paper could benefit from having a diagram of the conceptual model (i.e. last paragraph of the background section) that can help frame the paper.

The authors say “Finally, given that men are more likely to die from COVID and older adults are at higher risk of being infected by Sars-Cov-2, it is likely that gender and age could differentially impact adherence to social distancing behaviours.”. Can the author cite studies showing that men are more likely to die from COVID-19 and older adults are at higher risk of being infected by Sars-Cov-2? Also, I wonder if the fact that men die more than women and that older adults are more likely to be infected compared to their younger counterparts is the only reason why gender and age could impact adherence to social distancing behaviours differently. It would be good if the authors could expand on the other plausible reasons why we could find heterogeneous results. What does the literature tell about women behaving more cautiously than men, and why may adherence to preventative health behaviours vary by sex and age? Also, if men are more likely to die from COVID-19, they are likely to be more adherent to social distancing recommendations, but even the authors find that men are less adherent to social distancing recommendations.

The authors say “In the context of the COVID-19 pandemic, it seems reasonable to assume that individual reasons to adhere to social distancing measures (e.g., desire to protect self and others) as well as external circumstances or motivators (e.g., workplace/school conducted remotely) contribute to engagement in and adherence to preventative behaviours, such as social distancing. In addition, individual characteristics, such as demographic and psychological profile (educational level, health literacy, anxiety/stress, empathy towards others) might also play a role in adherence. Finally, given that men are more likely to die from COVID and older adults are at higher risk of being infected by Sars-Cov-2, it is likely that gender and age could differentially impact adherence to social distancing behaviours.”. One factor that is lacking in this paragraph is the family. An individual is part of a family. An individual may live with their partner, their kids, etc. They may also live with the most vulnerable people in this pandemic, such as their old parents or a partner with a pre-existing illness. The family composition may be an important socio-demographic predictor of social distancing behavioural outcomes. It would be good to include this factor in the analysis, or, if not available, at least discuss it in the background section.

Methods

The survey was piloted on 15 individuals whose data were not included in the analysis. It would be good to mention what if this is in line with what is usually done in the literature. Are surveys usually piloted on more/less than 15 individuals? And, what was their assessment of the survey, did they find it easy to complete?

I could not find the list of motivations for social distancing and social distancing behaviours in this section. The authors should here refer to Table 2 and Table 3 from the results section to allow the reader to know the motivations for social distancing and social distancing behaviours.

P. 9 The authors conceptualised adherence to social distancing as “always” endorsing the behaviour (coded as “1”) and nonadherence as behaviour endorsed less often than “always”, including “never”, “sometimes”, or “often” response choices (coded as “0”). It would be good to specify why “never” was treated the same way as “sometimes” and “often” and what this could imply for your results. If the reason is purely methodological, then I wonder why not using a tobit model. If conceptual, please specify. Also, please give an example of a behaviour where the “not applicable” option could be used.

Does the model include variables for country of residence? Because countries took different approaches (even within the same category ‘moderate rules’ / ‘strict rules’, there are differences in the measures adopted), the behaviours may also vary by country.

P. 9 The authors say: “During data collection, recommendations and policies for social distancing differed by region or country but did not change within one region or country, hence our regression models did not account for timing of survey completion.”. If, on the one hand, recommendations and policies did not change, on the other, the number of cases and deaths have increased over the period of analysis and this might have changed people’s behaviours by for example increasing their adherence to the social distancing measures. It would be good if the authors could account for the passage of time in their analysis.

Results

The results section is very difficult to follow because the results are presented as if they were reported on a presentation with bullet points. The whole section is organised in a similar fashion: “Endorsement rates for the four sets of motivations “for” (facilitators) and “against” (barriers) social distancing are included in Table 2. Highest endorsement rates were found for the following facilitators of social distancing: “I want to protect myself” (84%) and “I want to avoid spreading the virus to others” (83%) (individual-level facilitators); “I want to protect others” (86%) and “I feel a sense of responsibility to protect our community” (84%) (interpersonal-level); “My workplace/ school recommended we practice social distancing” (54%) and “My workplace /school conducts operations remotely” (51%) (organizational-level); “Restaurants in my area are closed for eating-in” (95%) and “Community centers and recreational facilities in my area are closed” (94%) (community-level).”. The authors should find a better way to present the results because the way it stands now is not ok.

The organizational-level motivations against social distancing stand out for having the lowest endorsement rates, i.e. “My workplace/ school recommended we practice social distancing” (54%) and “My workplace /school conducts operations remotely” (51%). Can the authors speculate why we get such low rates in this cluster?

Limitations

The implications of the limitations should be discussed in the paper. The authors mention three limitations of their study, but do not discuss their implications. Among them, the issue related to the sample selection is the most important. The authors cannot do much about it, and I think that they have been clear about the fact that the sample is not representative of the general population. On the other hand, I think the authors should at least discuss what are the implications of using this sample. If possible and sensible, I recommend having a table that compares the sample characteristics to the characteristics of the general population; this way we could at least know how the sample differs from the population.

Minor points

The authors say one of the social distancing measures is “maintaining a 2-metre distance between self and others when in public”. It would be good to specify what ‘when in public’ means. In particular, does social distancing apply to the private sphere too? For example, if I am visiting my parents at their home, do we still have to maintain the distance? Also, social distancing varies across countries from two metres down to one metre. It would be good to either be more general and say “at least a 1-metre distance”, or if the 2-metre rule is kept be more specific about where.

P.5 The authors say: “Since the World Health Organization (WHO) declared COVID-19 a pandemic on March 11, 2020, national and international public health agencies proposed several measures to contain or mitigate the virus transmission ranging from complete quarantine of the population of an entire region, as in Wuhan, China (virus containment) to various degrees of social distancing measures coupled with rigorous personal hygiene (e.g., washing hands frequently and thoroughly, avoiding touching the eyes, nose, and mouth, coughing and sneezing into the elbow; wearing face masks when in public) in Canada, the United States, and Europe (mitigation of transmission).”. The response was not the same across Europe, in fact some European governments imposed a national quarantine. On 9 March, i.e. two days before WHO announced COVID-19 outbreak a pandemic, the Italian government imposed a national quarantine like the Chinese government did in Wuhan. Italy was not the only one, others followed, e.g. Greece.

P.6 “Finally, given that men are more likely to die from COVID” should be “Finally, given that men are more likely to die from COVID-19”.

6. PLOS authors have the option to publish the peer review history of their article (what does this mean?). If published, this will include your full peer review and any attached files.

Reviewer #1: **Yes: **Liliana Andriano

---

## [Author Response · Author response to Decision Letter 0]

13 Aug 2020

Our response to the editor and reviewer's comments was uploaded as an attachment.

---

## [Decision Letter · Decision Letter 1]

4 Sep 2020

PONE-D-20-13845R1

Barriers and Facilitators of Adherence to Social Distancing Recommendations during COVID-19 among a Large International Sample of Adults

PLOS ONE

Dear Dr. Coroiu,

Thank you for submitting your manuscript to PLOS ONE. After careful consideration, we feel that it has merit but does not fully meet PLOS ONE’s publication criteria as it currently stands. Therefore, we invite you to submit a revised version of the manuscript that addresses the points raised during the review process.

We look forward to receiving your revised manuscript.

Kind regards,

Valerio Capraro

Academic Editor

PLOS ONE

Additional Editor Comments (if provided):

The reviewer suggests some additional minor changes. Please address these last points at your earliest convenience. Looking forward for the final version.

Reviewers' comments:

Reviewer's Responses to Questions

**Comments to the Author**

1. If the authors have adequately addressed your comments raised in a previous round of review and you feel that this manuscript is now acceptable for publication, you may indicate that here to bypass the “Comments to the Author” section, enter your conflict of interest statement in the “Confidential to Editor” section, and submit your "Accept" recommendation.

Reviewer #1: (No Response)

2. Is the manuscript technically sound, and do the data support the conclusions?

Reviewer #1: Yes

3. Has the statistical analysis been performed appropriately and rigorously? 

Reviewer #1: Yes

4. Have the authors made all data underlying the findings in their manuscript fully available?

Reviewer #1: Yes

5. Is the manuscript presented in an intelligible fashion and written in standard English?

Reviewer #1: Yes

6. Review Comments to the Author

Reviewer #1: (No Response)

7. PLOS authors have the option to publish the peer review history of their article (what does this mean?). If published, this will include your full peer review and any attached files.

Reviewer #1: **Yes: **Liliana Andriano

---

## [Author Response · Author response to Decision Letter 1]

11 Sep 2020

we attached the response to reviewers

---

## [Editor Report · Decision Letter 2]

15 Sep 2020

Barriers and Facilitators of Adherence to Social Distancing Recommendations during COVID-19 among a Large International Sample of Adults

PONE-D-20-13845R2

Dear Dr. Coroiu,

We’re pleased to inform you that your manuscript has been judged scientifically suitable for publication and will be formally accepted for publication once it meets all outstanding technical requirements.

Kind regards,

Valerio Capraro

Academic Editor

PLOS ONE
---

## [Editor Report · Acceptance letter]

24 Sep 2020

PONE-D-20-13845R2 

Barriers and Facilitators of Adherence to Social Distancing Recommendations during COVID-19 among a Large International Sample of Adults 

Dear Dr. Coroiu:

I'm pleased to inform you that your manuscript has been deemed suitable for publication in PLOS ONE. Congratulations! Your manuscript is now with our production department. 

Kind regards, 

on behalf of

Dr. Valerio Capraro 

Academic Editor

PLOS ONE